# The Comparison and Interpretation of Machine-Learning Models in Post-Stroke Functional Outcome Prediction

**DOI:** 10.3390/diagnostics11101784

**Published:** 2021-09-28

**Authors:** Shih-Chieh Chang, Chan-Lin Chu, Chih-Kuang Chen, Hsiang-Ning Chang, Alice M. K. Wong, Yueh-Peng Chen, Yu-Cheng Pei

**Affiliations:** 1Department of Physical Medicine and Rehabilitation, Chang Gung Memorial Hospital at Linkou, Taoyuan 333, Taiwan; kenny92031@cgmh.org.tw (S.-C.C.); lilychang0412@gmail.com (H.-N.C.); 2College of Medicine, Chang Gung University, Taoyuan 333, Taiwan; Dreamcheap2000@hotmail.com (C.-L.C.); leonard@cgmh.org.tw (C.-K.C.); 3Department of Neurology, New Taipei Municipal Tucheng Hospital, Chang Gung Memorial Hospital, New Taipei City 236, Taiwan; 4Department of Physical Medicine and Rehabilitation, Chang Gung Memorial Hospital at Taoyuan, Taoyuan 333, Taiwan; alicewong.mk@gmail.com; 5Healthy Aging Research Center, Chang Gung University, Taoyuan 333, Taiwan; 6Center for Artificial Intelligence in Medicine, Chang Gung Memorial Hospital at Linkou, Taoyuan 333, Taiwan; 7Center of Vascularized Tissue Allograft, Chang Gung Memorial Hospital at Linkou, Taoyuan 333, Taiwan

**Keywords:** machine learning, stroke, rehabilitation, post-acute care, functional recovery, activities of daily living

## Abstract

Prediction of post-stroke functional outcomes is crucial for allocating medical resources. In this study, a total of 577 patients were enrolled in the Post-Acute Care-Cerebrovascular Disease (PAC-CVD) program, and 77 predictors were collected at admission. The outcome was whether a patient could achieve a Barthel Index (BI) score of >60 upon discharge. Eight machine-learning (ML) methods were applied, and their results were integrated by stacking method. The area under the curve (AUC) of the eight ML models ranged from 0.83 to 0.887, with random forest, stacking, logistic regression, and support vector machine demonstrating superior performance. The feature importance analysis indicated that the initial Berg Balance Test (BBS-I), initial BI (BI-I), and initial Concise Chinese Aphasia Test (CCAT-I) were the top three predictors of BI scores at discharge. The partial dependence plot (PDP) and individual conditional expectation (ICE) plot indicated that the predictors’ ability to predict outcomes was the most pronounced within a specific value range (e.g., BBS-I < 40 and BI-I < 60). BI at discharge could be predicted by information collected at admission with the aid of various ML models, and the PDP and ICE plots indicated that the predictors could predict outcomes at a certain value range.

## 1. Introduction

Stroke is a major cause of disability and thus imposes substantial social and economic burdens [1,2]. Post-stroke rehabilitation is pivotal for managing disability and improving quality of life [3]. Because of the high diversity of stroke-induced disabilities, predicting their functional outcomes is difficult. Numerous factors may affect post-stroke functional outcomes, including age [4], cognition [5,6], comorbidities [7], post-stroke intervention [8,9], and stroke characteristics, such as severity [10], type [11], location [12,13], and volume [14]. Therefore, medical resources must be allocated to patients with a more favorable rehabilitation potential to help them achieve their rehabilitation goals.

Approximately 795,000 patients globally were newly diagnosed as having stroke in 2020 [15], and the cost of post-stroke care is expected to triple by 2035 [16]. Studies have mainly used regression models to define predictors for post-stroke functional outcomes. However, collinearity is a fundamental problem in this type of analysis because some variables might be excluded if they correlate with included predictors. To overcome this problem, novel prediction methods, such as machine learning (ML), that depend on advances in computational power have been developed.

An increasing number of ML models have been applied in the medical field. Unlike statistical models, which are used to determine the relationship between variables, ML models can be employed to make predictions using algorithms that can learn from data [17]. Several ML methods, such as support vector machine (SVM) [18], logistic regression [19], and decision trees [20], have been proposed to predict the prognostic outcomes of stroke. However, the method that yields the most satisfactory performance remains to be determined. Furthermore, to achieve superior predictive performance, ML models can be stacked using a meta-learning algorithm to combine predictions from multiple ML models [21]. Stroke is a complicated disease, and multiple factors can interfere with its outcomes. Therefore, a stacked ML model that incorporates the advantages of multiple ML models can yield a more favorable post-stroke outcome prediction.

The performance of an ML model largely depends on comprehensive variables. Data on patients enrolled in the Post-Acute Care-Cerebrovascular Disease (PAC-CVD) rehabilitation program, which was launched by Taiwan’s National Health Insurance (NHI) Administration in 2014, can be used to conduct ML model training. These data can be used to predict patients’ functional outcomes and allocate medical resources accordingly [22]. All eligible patients enrolled in the program under the NHI policy are transferred to accredited post-acute care (PAC) hospitals and receive an in-depth functional assessment at admission, interim, and discharge. The PAC program involves daily physical therapy (movement, balance, and ambulation), occupational therapy (activities of daily living), and speech and language therapy (communication and swallowing function) interventions for up to 12 weeks, which is nearly triple the rehabilitation period of a non-PAC program [22,23,24]. Thus, the data on patients enrolled in the PAC-CVD program are suitable for ML model training for post-stroke outcome prediction because of the high insurance coverage, in-depth functional assessment, and longer hospital stay.

Although ML models have been applied to predict post-stroke outcomes [18,19,20], the ML method that yields the most favorable performance remains to be determined. The number of predictors enrolled for model construction in these studies was limited and the relationship between predictors and predictive outcomes, such as feature importance, dependence, and heterogeneity, was not fully interpreted. In addition, studies have reported highly inconsistent results regarding post-stroke outcome prediction. Therefore, this study collected data regarding numerous predictors and applied multiple ML models to determine the model that demonstrates the most favorable predictive performance, identify predictors that are crucial for predicting post-stroke functional outcomes, and explore their performance in predicting functional outcomes.

## 2. Materials and Methods

### 2.1. Patients and Data Collection

Patients eligible for the PAC-CVD program conducted in Taoyuan Chang Gung Memorial Hospital were enrolled from March 2014 to December 2019. Information regarding 77 predictors was collected during acute ward stay and upon admission at a PAC hospital. The functional outcome was whether a patient could achieve a Barthel Index (BI) score of >60 upon discharge.

Upon admission to a PAC hospital, the patients’ demographic data, namely age and sex, were collected. In addition, patients were administered the following functional assessments (the “-I” represents the initial functional assessments): (1) the BI-I to examine functional independence; (2) the Lawton–Brody Instrumental Activities of Daily Living Scale (IADL-I) [25], modified Rankin Scale (mRS-I) [26], and European Quality of Life Five-Dimension Questionnaire to evaluate general disability; (4) the Berg Balance Scale (BBS-I) [27], Gait Speed-I [28], 6 min walk test (6-MWT-I) [29], Fugl–Meyer Upper Extremity Assessment (FuglUE-I), and modified Fugl–Meyer Sensory Assessment [30] to evaluate motor and sensory function; (5) the Functional Oral Intake Scale (FOIS-I) [31] to examine eating ability; (6) the Concise Chinese Aphasia Test (CCAT-I) to evaluate cognition and language function; and (7) the Mini-Nutritional Assessment (MNA-I) to evaluate nutritional status.

We retrospectively reviewed patients’ medical records in the acute management ward and collected the following data: (1) stroke characteristics, namely the subtype (hemorrhage vs infarction), territory (anterior cerebral artery or not), location (cortical, subcortical, or infratentorial), side (left, right, or bilateral), dissection, large-vessel stenosis, large-vessel occlusion, and undetermined; (2) post-stroke intervention, namely recombinant tissue plasminogen activator and intra-arterial thrombectomy; (3) the National Institute of Health Stroke Scale (NIHSS, comprising 15 subscales) score on admission and discharge in the acute ward [32]; (4) rehabilitation timing (the interval between the acute ward admission and commencement of rehabilitation); (5) comorbidities, namely hypertension, diabetes mellitus, dyslipidemia, atrial fibrillation, cardiovascular disease, chronic kidney disease, pulmonary disease, liver cirrhosis, hepatitis, malignancy, gout, parkinsonism, dementia, previous stroke, and psychiatric disorder; and (6) complications, namely pneumonia, urinary tract infection, stroke-in-evolution, gastrointestinal bleeding, and cellulitis. The length of stay (LOS, from admission to discharge in the acute ward) was recorded; however, this parameter was not included as a predictor.

The outcome was the BI at discharge (BI-F). The BI-F was further classified into BI-F > 60 and BI-F ≤ 60, which represented better and poorer outcomes, respectively.

### 2.2. The PAC-CVD Program

The patients were first evaluated by the case manager and enrolled in the PAC-CVD program if they met the following criteria: (1) enrollment within 1 month post-stroke, (2) stable hemodynamic parameters in the 72 h prior, (3) absence of neurological deterioration in the 72 h prior, and (4) adequate cognitive function and ability to participate in the rehabilitation program, with an mRS score of 2 to 4 (from 3 to 4 since July 2017 because of a change in the policy) [23,33]. Eligible patients were transferred to the PAC hospital and administered hospital-based multidisciplinary rehabilitation consisting of physical, occupational, and speech and language therapies. The patients underwent several functional assessments that were conducted by physiatrists and therapists at the beginning, interim, and end of the program. The team discussed the medical and functional progress, treatment plan, and rehabilitation goal on a regular basis [22].

The patients underwent three sessions (1 h per session) of tailored hospital-based multidisciplinary rehabilitation every weekday, comprising physical therapy (balance, gait, and robotic-assisted training) [34], occupational therapy (posture training, transfers, activity of daily living [ADL], cognitive training, and constraint-induced movement therapy) [35], and speech and language therapy (language and swallowing training).

Patients who possessed one or more of the following criteria were discharged from PAC-CVD hospital: (1) the ability to receive community-based rehabilitation for functional improvement, (2) absence of interval improvement on the basis of two consecutive functional evaluations, (3) absence of potential for functional recovery according to the evaluation of the care team, (4) a length of admission of more than 12 weeks, (5) a decision to terminate, and (6) death [22].

### 2.3. Ethics

The protocol of this study followed the Declaration of Helsinki and was approved by the Institutional Review Board (IRB) of Chang Gung Memorial Hospital (IRB No. 202101399B0, approved on 18 August 2021).

### 2.4. Eight ML Methods

A total of 77 candidate predictors were included. The outcome measurement of the ML models was BI-F > 60 (*n* = 397) or BI-F ≤ 60 (*n* = 180). The following eight ML methods were used: decision trees, Naïve Bayes, k-nearest neighbor (kNN), linear discriminant analysis, AdaBoost, SVM, logistic regression, and random forest (ensemble of bagged trees). The multilayer perceptron was not applied because a preliminary analysis indicated that its performance was inferior to those of other methods and that it was prone to being affected by hyperparameters (e.g., number of hidden layers and units and learning strategies); in addition, a proper architecture requires numerous computing resources [36]. To achieve superior predictive performance, we combined the results of each of the ML models using the stacking method to obtain an integrated result. The stacking method is an ensemble learning method that employs logistic regression to learn the ground truth from predicted scores generated by the eight ML models.

The models were constructed, and the analysis was performed using MATLAB’s Machine-Learning Toolbox and Statistics Toolbox (2021a release, MathWorks, Inc., Natick, MA, United States). Variable normalization techniques, such as the min–max normalization and standardization (except for Naïve Bayes, which requires the z-score transformation of data), and data reduction techniques, such as principal component analysis, were not applied because preliminary tests indicated that the use of these methods did not improve modeling performance.

### 2.5. Training and Validation

We applied the five-fold cross-validation method to develop the ML models. Following the five-fold cross-validation training procedure, before each training for each ML model, the 577 patients were randomly divided into two groups, the training and validation groups. Specifically, the training group that consisted of 80% (462 patients) of the dataset was applied for training the ML model while the remaining 20% (115 patients) for validating the trained ML model, an approach that could prevent overfitting. The aforementioned training procedure was repeated for 100 iterations. For each iteration, the patients in the dataset were randomly assigned into groups to avoid selection bias. After the completion of all the iterations, the performance of the ML model (e.g., AUC) was calculated over the 577 patients. Each type of ML model was constructed for 100 times and the performance of the 100 ML models were averaged to yield a summary of performance for each type of ML model.

During ML mode training, the tuning of hyperparameters was performed using Bayesian optimization with five-fold cross-validation. No data augmentation method was employed because the data were not heavily imbalanced (ratio of BI-F ≤ 60 to BI-F > 60 = 0.45:1).

### 2.6. Feature Importance Analysis

To compare the performance among the ML models, the entire procedure was repeated 20 times using randomly created five-fold datasets. To estimate the importance of each predictor [37], we used the permutation technique for the random forest model for the testing dataset. Using the permutation technique, we identified important features because the permutation of their values in the dataset would increase the model’s prediction error.

After defining the most important features, we constructed the individual conditional expectation (ICE) [38] plot and the partial dependent plot (PDP) [39] to characterize the contribution of each factor to the outcome measurement, namely the classification score. Each line of the ICE plot indicates the dependence of prediction on a specific predictor for each patient. By keeping the other predictors the same, the degree to which the prediction is affected by replacing the value of a predictor variable (shown on the x-axis) can be evaluated based on the changes in prediction values shown on the y-axis. The PDP plot is a summary of all lines in the ICE plot and represents the mean of prediction values (on the y-axis).

### 2.7. Statistical Analysis

Based on the receiver operating characteristic (ROC) curve of each ML model, the optimal cutoff point of the model output value for classification was determined by identifying the maximum of the Youden index (Youden index = sensitivity + specificity − 1). The performance of the ML models was evaluated by calculating the following indicators: area under the curve (AUC), $accuracy=\frac{TP+TN}{P+N}$, $specificity=\frac{TN}{FP+TN}$, and $sensitivity=\frac{TP}{TP+FN}$. A one-way analysis of variance (ANOVA) was performed to compare the difference in indicators among the ML models, and Tukey’s honestly significant difference (HSD) test was used for post hoc analysis. Pearson’s chi square test and the Wilcoxon rank-sum test were used to compare differences in demographic parameters, functional assessments, comorbidities, and complications between the BI-F > 60 and BI-F ≤ 60 groups. A *p* value of <0.05 was considered statistically significant.

## 3. Results

### 3.1. Patient Characteristics

A total of 577 patients were included in this analysis after 56 patients who did not undergo the assessment at the termination of the PAC-CVD program were excluded (acute ward readmission or against advice discharge). Post-stroke interventions were administered in 25 (4.3%) patients (intravenous thrombolysis, *n* = 16, and endovascular thrombectomy, *n* = 9). The mean rehabilitation timing, i.e., the interval from the stroke onset to the commencement of rehabilitation, was 13.2 ± 5.3 (mean ± standard deviation) days, and the LOS in the PAC-CVD ward was 52.3 ± 23.7 days. The mean BI scores upon admission to the PAC-CVD hospital and at discharge were 48.34 ± 16.9 and 71.2 ± 18.3, respectively.

In terms of demographic characteristics, all predictors exhibited no significant differences between the BI-F > 60 and BI-F ≤ 60 groups (all *p* > 0.05) except for age, rehabilitation timing, dyslipidemia, atrial fibrillation, hepatitis, malignancy, gout, previous stroke, and pneumonia. In terms of functional assessments, all predictors significantly differed between the BI-F > 60 and BI-F ≤ 60 groups (all *p* < 0.05) except for MNA-I (Table 1) and several NIHSS subscales.

After the completion of the PAC-CVD program, most (556, 96%) patients were discharged home, 2 (0.3%) developed acute conditions during PAC and were thus transferred to the acute ward, 2 (0.3%) were discharged against advice, 4 (0.6%) were transferred to a nursing home, and the remaining patients (13, 2.2%) were transferred to other rehabilitation facilities.

### 3.2. Performance of Post-Stroke Outcome Classification Models

The following eight ML models were trained: decision tree, Naïve Bayes, kNN, AdaBoost, linear discriminant analysis, SVM, logistic regression, and random forest. The stacking model was trained by applying the results of these eight models. The performance of the ML models was evaluated using ROC curves (Figure 1A). The AUC of these models ranged from 0.83 to 0.887, and the performance significantly differed among the ML models (one-way ANOVA, *p* < 0.001; Table 2). The results of the post hoc Tukey’s HSD test revealed that random forest, stacking, logistic regression, and SVM had comparable AUCs, indicating that the performance of these four models was equally superior to those of other ML models (Figure 1B).

### 3.3. Feature Importance for the Prognosis of BI at Discharge

The feature importance analysis of the 77 predictors indicated that BBS-I, BI-I, and CCAT-I were among the top three predictors (importance > 1 × 10^−3^). These predictors had lower values in the final BI ≤ 60 group, indicating that patients with lower BBS, BI, and CCAT scores at admission were more likely to experience difficulties in performing ADLs at discharge (Table 3). Figure 2 displays 20 of the predictors.

### 3.4. Dependence and Heterogeneity of Predictors

The dependence and heterogeneity of the top eight predictors were determined by creating the PDP and ICE plots (Figure 3). In the figures, the gray lines represent the ICE plot, indicating the functional relationship between the score of BI-F > 60 and the predictors, and the red line represents the PDP, indicating the mean of the lines in the ICE plot. Evaluating the PDP and ICE plots is crucial because heterogeneous effects might occur because of PDPs exhibiting only the average effects of ICE plots. For example, the PDP would be a horizontal line if half of the predictors had a positive association with the predictive outcome and the other half had a negative association. In this scenario, researchers might incorrectly conclude that the predictors did not affect the predictive outcome. Thus, by constructing the ICE plot, the heterogeneity can be evaluated.

These plots revealed that changes in the predictor value did not exhibit a linear relationship with changes in the predicted outcome (BI-F > 60). In the example of BBS-I (Figure 3A), the PDP (red line) and ICE (gray lines) plots sharply increased when the BBS-I value increased from 0 to 40, indicating that changes in the predicted outcome were mostly associated with BBS-I values ranging from 0 to approximately 40. Using the same rule, we discovered that BBS-I < 40, BI-I < 60, CCAT-I > 10, 6-MWT-I < 200 m, gait speed-I < 20 s, FuglUE-I < 60, mRS-I between 3 and 4, and IADL-I < 2 were more prone to yielding a change in the predicted outcome. These findings indicated that changes in predictor values at these ranges could account for the variability in ADL at discharge.

## 4. Discussion

The selection of appropriate ML models for disease prediction is crucial to optimize performance. The findings of this study indicated that the performance of random forest, stacking, logistic regression, and SVM models was superior to that of other ML methods. The stacking ML model, which combined the predictions of multiple ML models, did not demonstrate superior predictive performance, suggesting that any of the above-mentioned models exhibited the optimal performance in predicting post-stroke functional outcomes. Our study is unique in several ways. First, a total of 77 predictors were explored, including demographic parameters, functional assessments, comorbidities, and complications. Second, the PAC program enabled us to collect comprehensive and in-depth functional assessment data throughout the program. Third, the PAC program enabled patients to receive rehabilitation for up to 12 weeks, which is nearly triple the length of a non-PAC program.

The dependence and heterogeneity of the predictors were visually evaluated by constructing PDP and ICE plots. The ICE plot demonstrated the functional relationship between the outcome measurement (BI-F) and predictors, whereas the PDP represents the average of the lines of an ICE plot. Both plots enable investigators to determine the relationship between predictors and outcome measurements in a nonlinear manner. According to the PDP and ICE plots constructed in our study, changes in the outcome measurement were mostly associated with segmental changes in predictor values. These results can be explained by several aspects: (1) BBS-I < 40, BI-I < 60, 6-MWT-I < 200 m, and IADL-I < 2 were associated with changes in the BI-F, indicating that patients with poorer initial functional status tended to have more improvement and that the PAC-CVD program facilitated post-stroke functional improvement and (2) patients with CCAT-I > 10, gait speed-I < 20 s, FuglUE-I < 60, and mRS-I between 3 and 4 demonstrated considerable changes in BI-F scores possibly because of selection bias due to the stringent PAC-CVD inclusion criteria. This information provides valuable details for clinicians to assess patients and predict functional outcomes.

The results of the feature importance analysis revealed that the BBS-I, BI-I, and CCAT-I were the top three predictors. Patients with stroke often have impaired balance. One study reported a strong correlation between post-stroke functional abilities and initial balance function [40]. In addition, balance function is a prognostic factor for post-stroke recovery [41,42]. A study using data from Taiwan’s PAC-CVD database indicated that balance was the most significantly improved domain in stroke recovery [43]. Both static and dynamic balance are included in the BBS, thus representing the condition of trunk control and locomotion [27].

The BI-I is a strong prognostic factor for post-stroke recovery [44] and is considered the best predictor of the BI at discharge [45]. One study indicated that the BI-I and six other factors, namely age, diabetes, myocardial infarction, Brunnstrom motor recovery stages and motor control, time from stroke onset to acute care hospital admission, and time from admission to the commencement of rehabilitation, could explain up to 61% of the variance in discharge BI [45]. Thus, the BI-I can be used to predict post-stroke functional outcomes with simplicity and reliability.

Language function is crucial for rehabilitation and interpersonal relationships during recovery. Although gradual spontaneous recovery was observed in patients with aphasia in the first few months, a lifelong deficit remained in most patients [46,47,48]. Patients with aphasia are more likely to encounter difficulties in returning to work or participating in social activities, thus reducing their health-related quality of life [49]. Long-term recovery from aphasia can be predicted by the initial severity, size, and site of stroke [50]. Early identification of patients with aphasia and early intervention with speech and language therapy are crucial for increasing post-stroke independence.

Sex, comorbidities, and stroke-related complications were not identified as predictors in our ML models. However, this finding differs from those of other studies. Studies have reported that older women had a higher rate of stroke and poorer recovery [51], and poor recovery was independent of older age and other clinical or demographic characteristics [52]. Comorbidities, measured using the comorbidity index or weighted comorbidity index reported by Liu et al. [53], contributed 3% to the post-stroke functional outcome in one study [54]. The comorbidity index reported by Liu et al. comprises 41 conditions in the 13 main diagnostic categories (cardiovascular, pulmonary, orthopedics, metabolic and endocrine, gastrointestinal, neurologic and psychiatric, audiovisual, urologic, hematologic, infectious, neoplastic, dermatologic, and dental system) and has been widely used in studies on post-stroke outcome. These discrepancies in the findings may be related to the following aspects of this study: (1) selection bias caused by the strict admission criteria of the PAC-CVD program, (2) early timing of the outcome measurement, and (3) a small sample size.

This study demonstrated the power of ML methods that are less affected by the problem of collinearity [55] because the predictors were determined by performing a feature importance analysis. The PDP and ICE plots provided detailed information regarding the predictive values of the predictors for post-stroke functional outcome.

This study has several limitations. First, the participants were from northern Taiwan because they were enrolled from a PAC hospital in northern Taiwan. Second, patients who were considered to have poor potential for rehabilitation were excluded from the program because of the strict admission criteria of the PAC-CVD program. These factors may have caused selection bias. Third, the subtypes in ischemic strokes, such as lacunar infarction and cardioembolic stroke, were not further labelled in the present study. Among subtypes of ischemic stroke, lacunar infarction has the best functional prognosis [56], while patients with cardioembolic stroke have the poorest short-term prognosis [57]. Last, several factors, such as enriched environment [58], financial status [59], and nutritional status [60], were shown to affect the outcome of post-stroke rehabilitation. The data in the present study reflects the outcome of a hospital-based standard-of-care rehabilitation program that provides a comprehensive team care, including physical therapy, occupational therapy, speech therapy, and inpatient medical care. The knowledge in the present study can thus be applied as long as the hospitals or countries can provide a hospital-based standard-of-care rehabilitation program.

## 5. Conclusions

Information collected at admission could be used to predict patients’ ADL at discharge, measured by the BI, through various ML models. Random forest, stacking, logistic regression, and SVM demonstrated superior performance. The feature importance analysis identified the BBS-I, BI-I, and CCAT-I as the top three predictors. The PDP and ICE plots indicated that changes in the BI at discharge were mostly associated with a specific value of the predictors, indicating that these predictors were predictive at certain value ranges.

## Figures and Tables

**Figure 1 diagnostics-11-01784-f001:**
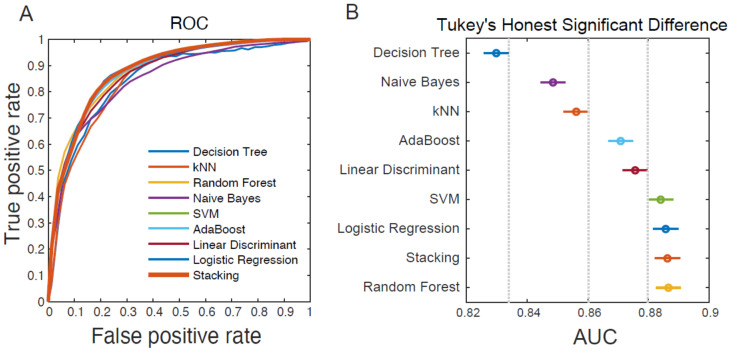
(**A**) Receiver operating characteristic (ROC) curves of different machine-learning models and performance group determined by (**B**) Tukey’s honestly significant difference test. The result indicates that random forest, stacking, logistic regression, and SVM are equally superior to those of other ML models. Data are represented as means ±95% confidence intervals. kNN: k-nearest neighbor; SVM: support vector machine.

**Figure 2 diagnostics-11-01784-f002:**
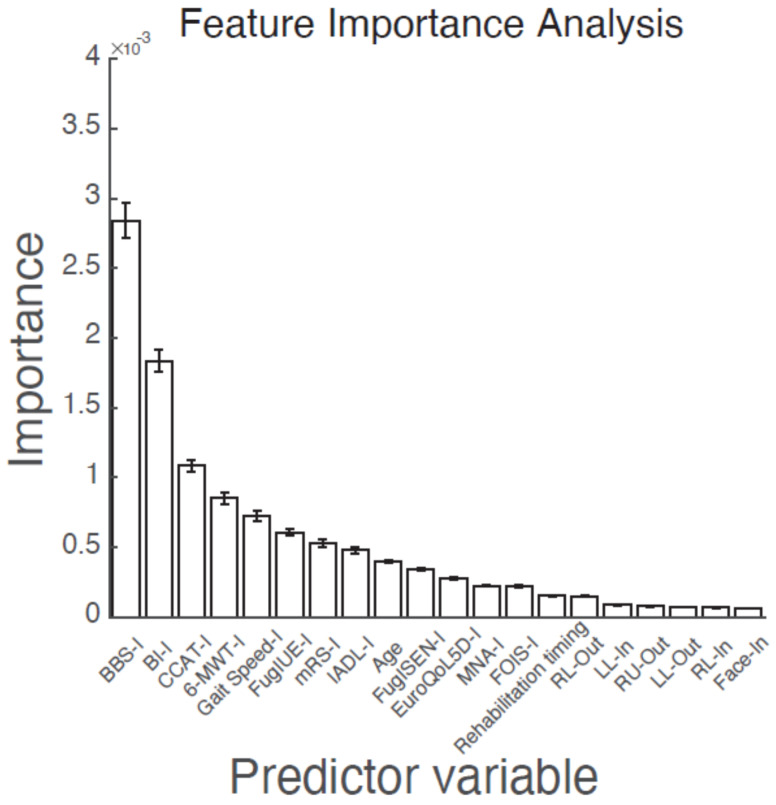
Feature importance of the random forest machine-learning model, indicating the 20 most important predictors. The “-I” represents the initial functional assessments at the PAC-CVD hospital. The “-In” and “-Out” represent the initial and final National Institute of Health Stroke Scale at the acute management ward. Error bars indicate the mean ± standard error. PAC-CVD: Post-Acute Care-Cerebrovascular Disease; BBS: Berg Balance Test; BI: Barthel Index; CCAT: Concise Chinese Aphasia Test; 6-MWT: 6-Minute Walk Test; FuglUE: Fugl–Meyer Upper Extremity Assessment; mRS: modified Rankin Scale; IADL: Lawton–Brody Instrumental Activities of Daily Living Scale; FuglSEN: modified Fugl–Meyer Sensory Assessment; Euro-QoL-5D: European Quality of Life Five-Dimension Questionnaire; MNA: Mini-Nutritional Assessment; FOIS: Functional Oral Intake Scale; LU: motor of the left arm; RU: motor of the right arm; LL: motor of the left Leg; RL: motor of the right Leg; Face: facial palsy.

**Figure 3 diagnostics-11-01784-f003:**
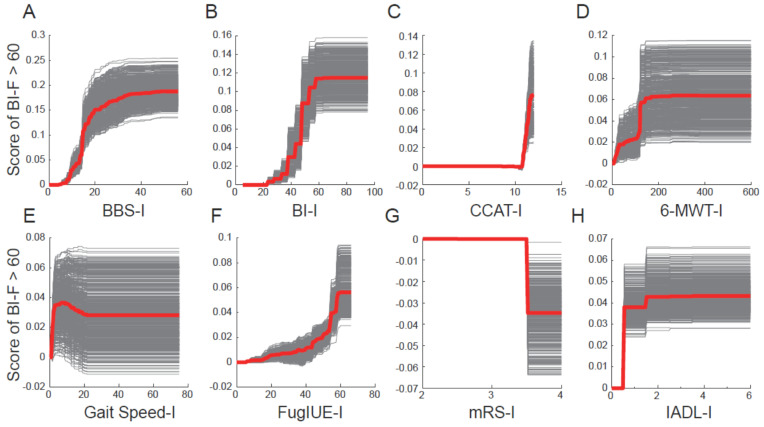
Partial dependence plot (PDP) and individual conditional expectation (ICE) plot. (**A**–**H**) BI-F > 60 as a function of BBS-I (**A**), BI-I (**B**), CCAT-I (**C**), 6-MWT-I (**D**), Gait Speed-I (**E**), FuglUE-I (**F**), mRS-I (**G**), and IADL-I (**H**). Gray lines represent the functional relationship between the score of BI-F > 60 and the predictors, and the red line represents the PDP and the mean of the ICE lines. A BI-F score of >60 is affected by the value of the above-mentioned parameters. The “-I” and “-F” represent the initial and final functional assessments at the PAC-CVD hospital. PAC-CVD: Post-Acute Care-Cerebrovascular Disease; BBS: Berg Balance Test; BI: Barthel Index; CCAT: Concise Chinese Aphasia Test; 6-MWT: 6-Minute Walk Test; FuglUE: Fugl–Meyer Upper Extremity Assessment; mRS: modified Rankin Scale; IADL: Lawton–Brody Instrumental Activities of Daily Living Scale.

**Table 1 diagnostics-11-01784-t001:** Patients’ demographic parameters.

Parameters	All	Group		
		Final BI > 60	Final BI ≤ 60	*p* Value
Patient number	577	397	180	
Age (year)	64.6 ± 12.6	62.4 ± 12.3	69.4 ± 11.8	<0.001 ***
Male/Female	381/196	271 (71)/126 (64)	110 (29)/70 (36)	0.093
Hemorrhagic stroke	84	61 (73)	23 (27)	0.414
Rehabilitation timing (day)	13.2 ± 5.3	12.6 ± 5.1	14.5 ± 5.7	<0.001 ***
BI-I	48.3 ± 16.9	54.7 ± 14.6	34.4 ± 12.6	<0.001 ***
BI-F	71.2 ± 18.3	81.1 ± 10.5	49.3 ± 11.4	<0.001 ***
ΔBI	22.9 ± 14.8	26.4 ± 14.9	15.0 ± 10.9	<0.001 ***
mRS-I	3.5 ± 0.6	3.4 ± 0.6	3.9 ± 0.4	<0.001 ***
FOIS-I	5.7 ± 1.9	6 ± 1.6	5.0 ± 2.2	<0.001 ***
MNA-I	16.5 ± 5.3	16.6 ± 5.5	16.3 ± 4.7	0.263
Euro-QoL-5D-I	9.8 ± 1.6	9.5 ± 1.5	10.4 ± 1.6	<0.001 ***
IADL-I	1.7 ± 1.2	1.9 ± 1.2	1.1 ± 1.1	<0.001 ***
BBS-I	29.3 ± 17.4	36 ± 14.4	14.5 ± 13.8	<0.001 ***
Gait speed-I (s)	6.6 ± 9.6	7.8 ± 9.4	4.0 ± 9.4	<0.001 ***
6-MWT-I (m)	115.1 ± 148.9	153.4 ± 157.0	30.6 ± 79.4	<0.001 ***
FuglUE-I	42.4 ± 20.0	47.2 ± 17.5	31.7 ± 20.8	<0.001 ***
FuglSEN-I	34.1 ±13.5	37.0 ± 10.7	27.9 ± 16.6	<0.001 ***
CCAT-I	10.7 ± 1.9	11.0 ± 1.49	9.9 ± 2.4	<0.001 ***
**Comorbidities**				
Hypertension	477 (77.5)	305 (76.8)	142 (78.9)	0.583
Diabetes mellitus	220 (38.1)	150 (37.8)	70 (38.9)	0.8
Dyslipidemia	271 (47.0)	199 (50.1)	72 (40.0)	0.024 *
Atrial fibrillation	54 (9.4)	28 (7.1)	26 (14.4)	0.005 **
Coronary arterial disease	49 (8.5)	31 (7.8)	18 (10.0)	0.382
Chronic kidney disease	25 (4.3)	13 (3.3)	12 (6.7)	0.064
Pulmonary disease	17 (2.9)	10 (2.5)	7 (3.9)	0.367
Liver cirrhosis	4 (0.7)	2 (0.5)	2 (1.1)	0.415
Hepatitis	15 (2.6)	14 (3.5)	1 (0.6)	0.038 *
Malignancy	26 (4.5)	13 (3.2)	13 (7.2)	0.034 *
Gout	38 (6.6)	33 (8.3)	5 (2.8)	0.013 *
Parkinsonism	8 (1.4)	4 (1.0)	4 (2.2)	0.248
Dementia	15 (2.6)	8 (2.0)	7 (3.9)	0.19
Old stroke	124 (21.5)	69 (17.4)	55 (30.6)	<0.001 ***
Psychiatric disorder	16 (2.8)	13 (3.3)	3 (1.7)	0.276
**Complications**				
Pneumonia	29 (5.0)	14 (3.5)	15 (8.3)	0.014 *
Urinary tract infection	36 (6.2)	20 (5.0)	16 (8.9)	0.076
Stroke-in-evolution	14 (2.4)	7 (1.8)	7 (3.9)	0.124
Gastrointestinal bleeding	16 (2.8)	10 (2.5)	6 (3.3)	0.581
Cellulitis	10 (1.7)	7 (1.8)	3 (1.7)	0.934

Values are expressed as means ± standard deviations or counts (percentages). The “-I” and “-F” represent the initial and final functional assessments at the PAC-CVD hospital. The “-In” and “-Out” represent the initial and final National Institute of Health Stroke Scale at the acute management ward. PAC-CVD: Post-Acute Care-Cerebrovascular Disease; BI: Barthel Index; BBS: Berg Balance Test; ΔBI: change from the initial BI to the final BI; CCAT: Concise Chinese Aphasia Test; mRS: modified Rankin Scale; IADL: Lawton–Brody Instrumental Activities of Daily Living Scale; Euro-QoL-5D: European Quality of Life Five-Dimension Questionnaire; 6-MWT: 6-Minute Walk Test; FuglUE: Fugl–Meyer Upper Extremity Assessment; FuglSEN: modified Fugl–Meyer Sensory Assessment; FOIS: Functional Oral Intake Scale; MNA: Mini-Nutritional Assessment. * *p* < 0.05; ** *p* < 0.01; *** *p* < 0.001.

**Table 2 diagnostics-11-01784-t002:** Performance of machine-learning models.

Model	AUC	ACC	Spe ^†^	Sen ^†^
Decision Tree (a)	0.83 ± 0.048	0.817 ± 0.009	0.749 ± 0.067	0.828 ± 0.056
Naïve Bayes (b)	0.849 ± 0.008	0.786 ± 0.005	0.811 ± 0.074	0.744 ± 0.075
kNN (c)	0.856 ± 0.006	0.828 ± 0.006	0.709 ± 0.042	0.866 ± 0.041
AdaBoost (d)	0.871 ± 0.025	0.827 ± 0.011	0.792 ± 0.042	0.83 ± 0.04
Linear Discriminant (e)	0.876 ± 0.008	0.819 ± 0.009	0.785 ± 0.045	0.813 ± 0.046
SVM (f)	0.884 ± 0.003	0.831 ± 0.005	0.791 ± 0.022	0.841 ± 0.024
Logistic Regression (g)	0.886 ± 0.003	0.833 ± 0.005	0.794 ± 0.015	0.85 ± 0.041
Stacking (h)	0.886 ± 0.005	0.831 ± 0.005	0.859 ± 0.005	0.758 ± 0.011
Random Forest (i)	0.887 ± 0.003	0.829 ± 0.005	0.781 ± 0.036	0.829 ± 0.033
*p* value	<0.001 *** a; b,c; d,e; f,g,h,i			

Values are expressed as means ± standard deviations. kNN: K-nearest neighbor; SVM: support vector machine; AUC: area under the curve; ACC: accuracy; Spe: specificity; Sen: sensitivity. † Sensitivity and specificity were computed using the maximum of Youden’s Index as the cutoff value. ***: *p* < 0.001.

**Table 3 diagnostics-11-01784-t003:** The top three importance predictors in the random forest machine-learning model to predict BI-F.

Parameter	Final BI > 60	Final BI ≤ 60	*p* Value
BBS-I	36.0 ± 14.4	14.5 ± 13.8	<0.001 **
BI-I	54.7 ± 14.6	34.4 ± 12.6	<0.001 **
CCAT-I	11.0 ± 1.49	9.9 ± 2.4	<0.001 **

Values are expressed as means ± standard deviations. The “-I” represents the initial functional assessments at the PAC-CVD hospital. PAC-CVD: Post-Acute Care-Cerebrovascular Disease; BBS: Berg Balance Test; BI: Barthel Index; CCAT: Concise Chinese Aphasia Test. ** *p* < 0.001.

## Data Availability

Data available on request due to restrictions, e.g., privacy or ethical.

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
