# Peer review of "The Comparison and Interpretation of Machine-Learning Models in Post-Stroke Functional Outcome Prediction"

_diagnostics, 2021, doi:10.3390/diagnostics11101784_

Round 1
Reviewer 1 Report
New prediction methods, such as machine learning (ML), that rely on advances in computational power, have been developed to better define predictors of functional outcomes after stroke. The authors of the present study collected data on admission from 577 stroke patients regarding a total of 77 predictors (demographic parameters, functional assessments, comorbidities, and complications) that were explored and had multiple ML models applied to them to determine the model demonstrating the most favorable predictive performance at discharge, identify predictors that are crucial for predicting post-stroke functional outcomes, and explore their performance in predicting functional outcomes. The authors found that the initial Berg Balance Test (BBS-I), initial Barthel Index (BI), and initial Concise Chinese Aphasia Test (CCAT-I) were the top three predictors of BI scores >60 at discharge, indicating that patients with lower BBS, BI, and CCAT scores at admission were more likely to experience difficulties in performing ADLs at hospital discharge. The study is potentially interesting, but can be improved if the following considerations are addressed:
-1. it would be interesting to include a comment that lacunar infarcts are the ischemic stroke subtype with the best functional prognosis (J Neurol Neurosurg Psychiatry 2001; 71: 239-242). Please refer to and include this supporting reference. It would be interesting to know the frequency of lacunar stroke in the study sample.
-2. It would be interesting to know the different stroke subtypes in the study population.
-3. It would be helpful to mention that in-hospital mortality rate on cardioembolic stroke remains around 20% and cardioembolic stroke is the subtype of ischemic infarct with the highest in-hospital mortality. The short-term prognosis of patients with cardioembolic stroke is poor compared to other ischemic stroke subtypes (Curr Cardiol Rev 2010; 6: 150-161)
-4.Check references #5 and #28
Reviewer 2 Report
Thank you for the opportunity to review the paper that predicted the effect of PAC-CVD on patients with stroke using machine learning techniques. This manuscript could be used as a reference for setting goals for patients with stroke, as a result of obtaining accurate prediction methods through carefully planned calculations. I think this manuscript can be published in the Diagnostics, I hope my minor comments will be helpful in revising this manuscript.
Comment 1: The authors wrote in the manuscript that 577 subjects were divided into 20:80 by the five-fold cross-validation method for machine learning and tried 100 times. Which of the divided groups used as training data? Also, was the identity of these 100 datasets maintained? Or did authors integrate 100 trials to calculate ROC and AUC? For example, reviewers could not read how the SVM training and test data were set.
Comment 2: Comparing Table 1 and Tabel 3, patients with Final BI> 60 were recognized as cut-off values for pre-intervention BI, BBS, and CCAT to predict ADL independence. Was the Initial data in Table 1 the same as the other predictions? Tabel 3 shows the parameters in the stacking model, but I wasn't sure if they were the same in other prediction models. If the authors clarify this point, it will be easier for readers to cite it.
Comment 3: For the male parameter of Tabel 1, why not describe the female in parentheses in All?
Comment 4: Is the extent to which the results of this study can be referred to limited to the Taiwan region? Globally, it is speculated that the outcome of rehabilitation may be influenced by the home environment, the nutritional status of patients, and the financial status, but this was not mentioned in this study.
